# Synthesis of Silane-Based Poly(thioether) via Successive Click Reaction and Their Applications in Ion Detection and Cell Imaging

**DOI:** 10.3390/polym11081235

**Published:** 2019-07-25

**Authors:** Zhiming Gou, Xiaomei Zhang, Yujing Zuo, Weiying Lin

**Affiliations:** Institute of Fluorescent Probes for Biological Imaging, School of Chemistry and Chemical Engineering, School of Materials Science and Engineering, University of Jinan, Shandong 250022, China

**Keywords:** poly(thioether), ion detection, cell imaging, organosilicone, fluorescence quenching

## Abstract

A series of poly(thioether)s containing silicon atom with unconventional fluorescence were synthesized via successive thiol click reaction at room temperature. Although rigid π-conjugated structure did not exist in the polymer chain, the poly(thioether)s exhibited excellent fluorescent properties in solutions and showed visible blue fluorescence in living cells. The strong blue fluorescence can be attributed to the aggregation of lone pair electron of heteroatom and coordination between heteroatom and Si atom. In addition, the responsiveness of poly(thioether) to metal ions suggested that the selectivity of poly(thioether) to Fe^3+^ ion could be enhanced by end-modifying with different sulfhydryl compounds. This study further explored their application in cell imaging and studied their responsiveness to Fe^3+^ in living cells. It is expected that the described synthetic route could be extended to synthesize novel poly(thioether)s with superior optical properties. Their application in cell imaging and ion detection will broaden the range of application of poly(thioether)s.

## 1. Introduction

Poly(thioether) are important base polymers that exhibit excellent physical and chemical properties, such as good chemical and weather resistance, high impermeability, and high heat insulation properties [1]. These excellent properties made it to become a promising functional polymer for the application of optical materials, medical impression materials ion detection probes, energy storage and conversion materials [2,3,4]. Thus far, many methods have been developed for the ion detection, including microelectrode, absorption spectroscopy [5]. However, these methods are difficult to dynamically monitor cellular ion changes. By contrast, fluorescence spectroscopy has become a powerful tool for fluorescent labeling, ion sensing and cell imaging due to its high sensitivity, excellent selectivity and dynamic monitoring [6,7]. However, as one kind of functional polymer, poly(thioether)s are rarely applied in cell imaging, especially for Fe ion detection in living cells.

At present, two synthetic routes were adopted to prepare poly(thioether). One is the ring-opening polymerization of episulfide [8,9], and the other is thiol click polymerization between dithiols and dienes or dithiols and alkyne [10,11,12]. As one of most widely used and important reactions, the thiol click reaction has served as useful tool to fabricated dendrimer and block polymer [13,14], and also has been used for end-/side-group functionalization and surface modification [15,16].

Trimethylsilylacetylene (TMSA) is a versatile precursor to prepare important intermediates or unsaturated compounds with unique properties, and also to be used as a protecting group for functional modification. NaKa group and Miura group reported the synthesis of silyl-substituted thiophene derivatives and silyl-substituted fulvene derivatives based on TMSA, respectively [17,18]. Yokozawa et al. described the synthesis of ethynyl-functionalized poly(3-hexylthiophene) with one terminal ethynyl group, which has been very suitable for subsequent preparation of block copolymers via a click reaction [19]. Grirrane et al. synthesized a novel family of dipropargylamines through a double catalytic A^3^-coupling of primary amine, formaldehyde and TMSA, and subsequent deprotection of two terminal trimethylsilyl (TMS) groups endow them with high values of post-modification [20]. Furthermore, Alejandro and Pilar reported metal-monolayer-metal molecular electronic devices based on oligoynes (Me_3_Si–(C=C)_4_–SiMe_3_) through fluoride-induced deprotection of terminal trimethylsilyl (TMS) groups [21,22], and this nascent surface modification technique provided new perspectives for the fabrication of molecular electrochemical sensors.

Among the above reports, studies on poly(thioether)s using TMSA are rare. Here, this study reported the synthesis of functionalized poly(thioether)s containing silane via a two-step successive addition reaction (Scheme 1). Firstly, the addition of trimethylsilylacetylene (TMSA) to a dithiol with different molar ratio yielded a series of poly(thioether)s. Secondly, vinyl poly(thioether) (**P1** and **P2**) was selected as intermediates, and reserved vinyl groups further reacted with thiol and finally gained the functionalized poly(thioether)s. The authors further explored their fluorescence properties and used them in cell imaging and ion detection. It was found that the post-functionalization of poly(thioether)s obviously enhanced their responsiveness to Fe^3+^ and avoided the disturbance of Fe^2+^. The obtained poly(thioether)s and post-functionalized products provided a simple route to synthesize sulfur-containing organosilicone polymer with superior optical properties and extended their application in the field of cell imaging and ion detection.

## 2. Materials and Experiments

### 2.1. Materials

Trimethylsilylacetylene (TMSA), 2,2-dimethoxy-2-phenylacetophenone (DMPA), 1,2-ethanedithiol (EDT), and 3,6-dioxa-1,8-octanedithiol (DODT) were purchased from Aladdin Co. (Shanghai, China) and used directly. Furfuryl mercaptan, trimethoxysilylpropanethiol, and methyl 3-mercaptopropionate were supplied by Macklin biochemical (Shanghai, China) Co., Ltd. and used as received. All solvents were bought from Chemical Technology (Shanghai, China) Co., Ltd. and directly used without further purification.

### 2.2. Characterization and Measurements

NMR (^1^H and ^13^C) spectra were recorded by Bruker AVANCE 400 MHz using CDCl_3_ as solvents, and without tetramethylsilane as the internal standard. Thermogravimetric analysis (TGA) was measured using SDTQ 600 at 10 °C·min^−1^ under N_2_. The luminescence emission spectra of the poly(thioether)s were recorded by using a Hitachi F–4500 fluorescence spectrophotometer with Xe lamp as an excitation source. The data of molecular weights was measured by gel permeation chromatography (GPC) using a Waters 515 liquid chromatograph with a refractive index detector 2414 and using tetrahydrofuran (THF) as the elution solvent.

### 2.3. Synthesis of Poly(thioether)

The synthetic route and structure of poly(thioether)s have been shown in Scheme 1. The detailed procedure for **P1** was as follows: TMSA (0.98 g, 10.0 mmol), EDT (0.75 g, 8.0 mmol) and DMPA (1 wt%) were placed in a glass vessel. The above mixture was exposed to a UV light source (365 nm, 100 w) with stirring for 20 min at room temperature. The crude product was washed three times with cold methanol and then reduced pressure distillation. The obtained light-yellow solid was **P1**. Yield: 96%. A series of poly(thioether)s (**P2**, **P3**, **P4**, **P5**, and **P6**) were synthesized according to the same above reaction condition and the treating process.

The data of **P1**: ^1^H-NMR (400 MHz, CDCl_3_, ppm): δ = 0.05–0.20 (-SiC*H*_3_), 2.05 (-SiC*H*CH_2_S-), 2.69–3.14 (-SiCHC*H*_2_S-, -SC*H*_2_C*H*_2_S-), 5.72 and 5.77 (s, -SiC*H*_2_=CH_2_S-), 6.47 and 6.52 (s, -SiCH_2_=C*H*_2_S-). ^13^C-NMR (400 MHz, CDCl_3_, ppm): δ = −3.0 (Si*C*H_3_), 31.6 (-SiCH*C*H_2_S-), 32.8 and 36.5 (-S*C*H_2_*C*H_2_S-, -Si*C*HCH_2_S-), 110.7 and 133.0 (-Si*C*H_2_=*C*H_2_S-).

The data of **P2**: ^1^H-NMR (400 MHz, CDCl_3_, ppm): δ = 0.07–0.13 (-SiC*H*_3_), 2.07 (-SiC*H*CH_2_S-), 2.75–2.95 (-SiCHC*H*_2_S-, -SC*H*_2_CH_2_O-), 3.03–3.08 (-OC*H*_2_C*H*_2_O-), 3.61–3.70 (-SCH_2_C*H*_2_O-), 6.49 and 6.53 (-SiCH_2_=C*H*_2_S-). ^13^C-NMR (400 MHz, CDCl_3_, ppm): 2.06–2.07 (Si*C*H_3_), 29.7 and 32.89 (-OCH_2_*C*H_2_S-), 31.7 (-SiCH*C*H_2_S-), 36.7 (-Si*C*HCH_2_S-), 70.7 (-O*C*H_2_*C*H_2_O-), 70.9 (-SCH_2_*C*H_2_O-), 110.9 and 133.1 (-Si*C*H_2_=*C*H_2_S-). Yield: 95%.

The data of **P3**: ^1^H-NMR (400 MHz, CDCl_3_, ppm): δ = 0.06–0.19 (m, -SiC*H*_3_), 2.07 (-SiC*H*CH_2_S-), 2.79–3.09 (-SiCHC*H*_2_S-, -SC*H*_2_C*H*_2_S-). ^13^C-NMR (400 MHz, CDCl_3_, ppm): −3.0 (Si*C*H_3_), 24.6 (-SCH_2_*C*H_2_SH), 31.6 (-SiCH*C*H_2_S-), 32.9 (-S*C*H_2_*C*H_2_S-), 33.7 (-S*C*H_2_CH_2_SH), 36.5 (-Si*C*HCH_2_S-). Yield: 97%.

The data of **P4**: ^1^H-NMR (400 MHz, CDCl_3_, ppm): δ = 0.07–0.13 (-SiC*H*_3_), 2.07 (-SiC*H*CH_2_S-), 2.73–2.91 (-SiCHC*H*_2_S-, -SC*H*_2_CH_2_O-), 3.02–3.06 (-OC*H*_2_C*H*_2_O-), 3.62–3.68 (-SCH_2_C*H*_2_O-). ^13^C-NMR (400 MHz, CDCl_3_, ppm): 2.06–2.07 (Si*C*H_3_), 29.8 and 32.9 (-OCH_2_*C*H_2_S-), 31.8 (-SiCH*C*H_2_S-), 36.9 (-Si*C*HCH_2_S-), 70.8 (-O*C*H_2_*C*H_2_O-), 70.9 (-SCH_2_*C*H_2_O-). Yield: 93%.

The data of **P5**: ^1^H-NMR (400 MHz, CDCl_3_, ppm): δ = 0.05–0.16 (m, -SiC*H*_3_), 1.35 (-SCH_2_CH_2_S*H*), 2.05 (-SiC*H*CH_2_S-), 2.69–3.05 (-SiCHC*H*_2_S-, -SC*H*_2_C*H*_2_S-). ^13^C-NMR (400 MHz, CDCl_3_, ppm): −1.76 (Si*C*H_3_), 24.6 (-SCH_2_*C*H_2_SH), 31.6 (-SiCH*C*H_2_S-), 32.9 (-S*C*H_2_*C*H_2_S-), 33.7 (-S*C*H_2_CH_2_SH), 36.5 (-Si*C*HCH_2_S-). Yield: 92%.

The data of **P6**: ^1^H-NMR (400 MHz, CDCl_3_, ppm): δ = 0.07–0.16 (-SiC*H*_3_), 1.34 (-OCH_2_CH_2_S*H*), 2.09 (-SiC*H*CH_2_S-), 2.74–3.07 (-SiCHC*H*_2_S-, -SC*H*_2_CH_2_O-), 3.62–3.64 (-OC*H*_2_C*H*_2_O-), 3.70–3.75 (-SCH_2_C*H*_2_O-, -OC*H*_2_CH_2_SH). ^13^C-NMR (400 MHz, CDCl_3_, ppm): 2.07 (Si*C*H_3_), 21.0 (-OCH_2_*C*H_2_SH), 29.6 and 32.9 (-OCH_2_*C*H_2_S-), 67.1–71.3 (-O*C*H_2_*C*H_2_O-, -SCH_2_*C*H_2_O-, -O*C*H_2_CH_2_SH). Yield: 94%.

Methyl-mercaptopropionate, trimethoxysilylpropanethiol and furfuryl mercaptan were selected and used to the functionalization of **P1** and **P2**. Taking methyl-mercaptopropionate as an example, the above obtained **P1**, methyl-mercaptopropionate (4.0 mmol) and DMPA (1 wt%) were dissolved in THF (2 mL) and reacted following the above-mentioned procedure. The crude product was washed three times with cold methanol and then reduced pressure distillation. The obtained light-yellow solid was **P1-1**. Yield: 93%. A series of functionalized poly(thioether)s (**P1-1**, **P1-2**, **P1-3**, **P2-1**, **P2-2**, and **P2-3**) were synthesized according to the same procedure.

The data of **P1-1**: ^1^H NMR (400 MHz, CDCl_3_): 0.03–0.09 (SiC*H_3_*), 2.21 (SiC*H*CH_2_S), 2.47–2.51(SCH_2_C*H*_2_COOCH_3_), 2.60–2.99 (SiCHC*H*_2_S, SC*H*_2_C*H*_2_S), 3.61 (COOC*H**_3_*). ^13^C-NMR (400 MHz, CDCl_3_, ppm): −3.01 (Si*C*H*_3_*), 29.6 (S*C*H_2_CH_2_COCH_3_), 31.61 (S*C*H_2_CH_2_O), 32.7 (SiCH*C*H_2_S), 33.4 (SCH_2_*C*H_2_COCH_3_), 36.5 and 38.1 (Si*C*HCH_2_S), 69.8 (O*C*H_2_CH_2_S), 172.1 (*C*OOCH_3_).

The data of **P1-2**: ^1^H NMR (400 MHz, CDCl_3_): 0.01–0.10 (SiC*H_3_*), 2.23 (SiC*H*CH_2_S), 2.51 and 3.18 (SiCHC*H*_2_S), 2.58–3.01 (SC*H*_2_C*H*_2_S), 3.33–3.43 (SC*H_2_*CO), 6.27 and 6.40 (C=C*H_2_*C*H_2_*CH*_2_*O), 7.58 (C=CH_2_CH_2_C*H**_2_*O). ^13^C NMR (400 MHz, CDCl_3_): −1.75 (SiC*H_3_*), 30.0 (S*C*H_2_CO), 31.0 and 31.6 (S*C*H_2_CO), 32.6 and 34.6 (SiCH*C*H_2_S), 36.4(Si*C*HCH_2_S), 106.9–111.1 (-C=*C*H*C*H=*C**H*O-), 142.3 (-C=CHCH=C*H*O-), 151.4 (-*C*=C*H*C*H=*CHO-). Yield: 94%.

The data of **P1-3**: ^1^H NMR (400 MHz, CDCl_3_): 0.01–0.15 (SiC*H_3_*), 0.56–0.74 (OSiC*H**_2_*CH_2_CH_2_S), 1.53–1.66 (OSiCH_2_C*H**_2_*CH_2_S), 2.17–2.26 (SiC*H*CH_2_S), 2.43-2.52 and 3.18 (SiCHC*H*_2_S), 2.63–3.04 (SC*H*_2_C*H*_2_S), 3.41–3.50 (OC*H**_3_*). ^13^C NMR (400 MHz, CDCl_3_): −2.0 (SiC*H_3_*), 8.35 (Si*C*H_2_CH_2_CH_2_S), 9.65 (SiCH_2_*C*H_2_CH_2_S), 31.4 and 31.7 (S*C*H_2_*C*H_2_S, SiCH*C*H_2_S), 33.1 (SiCH_2_CH_2_*C*H_2_S), 36.7 (Si*C*HCH_2_S), 48.9 and 50.2 (O*C*H_3_). Yield: 93%.

The data of **P2-1**: ^1^H NMR (400 MHz, CDCl_3_): 0.07–0.10 (SiC*H_3_*), 2.23 (SiC*H*CH_2_S), 2.59–2.65 (SC*H**_2_*CH_2_O), 2.66–2.76 (SiCHC*H*_2_S), 2.78–2.85 (SCH_2_C*H*_2_CO,), 3.52 (OC*H*_2_C*H*_2_O), 3.54–3.60 (SCH_2_C*H**_2_*O), 3.61 (OC*H**_3_*). ^13^C-NMR (400 MHz, CDCl_3_, ppm): −1.89 and −0.53 (Si*C*H*_3_*), 29.8 (S*C*H_2_CH_2_COCH_3_), 31.61 (S*C*H_2_CH_2_O), 32.9 (SiCH*C*H_2_S), 33.6 (SCH_2_*C*H_2_COCH_3_), 36.7 and 38.3 (Si*C*HCH_2_S), 51.9 (COO*C*H_3_), 69.8 (O*C*H_2_*C*H_2_O), 70.9 (O*C*H_2_CH_2_S), 172.3 (*C*OOCH_3_). Yield: 92%.

The data of **P2-2**: ^1^H NMR (400 MHz, CDCl_3_): 0.01–0.09 (SiC*H_3_*), 2.24 (SiC*H*CH_2_S), 2.67–2.76 (SC*H**_2_*CH_2_O), 2.51 and 2.76–2.92 (SiCHC*H*_2_S), 3.52–3.54 (SC*H_2_*CO), 3.55–3.59 (OC*H**_2_*C*H**_2_*O), 3.75–3.95 (SC*H**_2_*C*H**_2_*O), 6.26 and 6.38 (-C=C*H*C*H=*CHO-), 7.57 (-C=CHCH=C*H*O-). ^13^C NMR (400 MHz, CDCl_3_): −1.74 (SiC*H_3_*), 29.7 (S*C*H_2_CO), 31.0 and 31.8 (S*C*H_2_CH_2_O), 32.8 and 34.7 (SiCH*C*H_2_S), 36.6(Si*C*HCH_2_S), 70.0 (O*C*H_2_*C*H_2_O), 70.8 (SCH_2_*C*H_2_O), 106.9–111.0 (-C=*C*H*C*H=*C**H*O-), 142.7 (-C=CHCH=C*H*O-), 151.8 (-*C*=C*H*C*H=*CHO-). Yield: 94%.

The data of **P2-3**: ^1^H NMR (400 MHz, CDCl_3_): 0.07–0.10 (SiC*H_3_*), 0.58–0.72 (OSiC*H**_2_*CH_2_CH_2_S), 1.55–1.62 (SiCH_2_C*H**_2_*CH_2_S), 2.24 (SiC*H*CH_2_S), 2.43–2.52 and 2.99 (SiCHC*H*_2_S), 3.42 (SC*H_2_*CO), 3.48–3.59 (OC*H**_2_*C*H**_2_*O), 3.50–3.54 (OC*H**_3_*), 3.42–3.59 (SCH_2_C*H**_2_*O). ^13^C NMR (400 MHz, CDCl_3_): −1.9 (SiC*H_3_*), 8.33 (Si*C*H_2_CH_2_CH_2_S), 9.67 (SiCH_2_*C*H_2_CH_2_S), 31.3 and 31.7 (S*C*H_2_CH_2_O, SiCH*C*H_2_S), 32.9 (SiCH_2_CH_2_*C*H_2_S), 36.7 (Si*C*HCH_2_S), 48.9 and 50.1 (O*C*H_3_), 70.0 (O*C*H_2_*C*H_2_O), 70.9 (SCH_2_*C*H_2_O). Yield: 91%.

### 2.4. Fluorescence Property and Ion Detection

Poly(thioether)s were dissolved in ethanol (0.3 mg/mL) and their fluorescent emission spectra were measured under excitation wavelengths of 330 nm.

The responsiveness of poly(thioether)s to ions were detected by adding corresponding chlorine salt (100 μL, 1 mmol/L) to poly(thioether)s solution (900 μL, 0.2 mg/mL).

The HeLa cells were incubated with 4.0 μg of poly(thioether)s for 20 min at 37 °C and washed with phosphate buffer solution (PBS). The cell images of poly(thioether)s in HeLa cells were observed under a confocal microscope after adding new culture medium. In addition, the responsiveness of poly(thioether)s to Fe^3+^ in HeLa cells were measured by successively adding 10 μL, 30 μL, 50 μL and 70 μL Fe^3+^ ion (1 mmol/L), respectively. The response process was observed in situ under confocal microscopy.

## 3. Results and Discussion

### 3.1. Synthesis of Poly(thioether)s and Post-Functionalization

A series of poly(thioether)s were synthesized using different molar ratio (1:0.8, 1:1.0, and 1:1.2) between TMSA and dithiol (Scheme 1). In view of the same reaction condition and purification step, 1,2-ethanedithiol based poly(thioether)s (**P1**, **P3**, and **P5**) was selected to be analyzed and their ^1^H NMR have been shown in Figure 1. The characteristic peak of alkenyl groups (-C*H*=C*H-*) appeared at approximately 5.75 ppm and 6.5 ppm only existed in the polymer when the molar ratio was 1:0.8. By the same token, the characteristic peak of sulfhydryl groups (-S*H*) appeared at approximately 1.7 ppm and was presented when the molar ratio was 1:1.2. The appearance of the peak at 1.7 ppm when the molar ratio was 1:1 could be attribute to the low boiling point of TMSA, which resulted in the volatilization of part of TMSA in the reaction process. In addition, the thiol-alkyne reaction proceeded under a radical step-growth mechanism involving a two step addition. Therefore, the dithiol radicals can react with alkynyl groups from different active sites and generate two types of product structures (α-addition and β-addition) which exist simultaneously in the polymer backbone. The appearance of peak at 1.35 ppm belonged to α-addition products. However, a small amount of α-addition could not have had a significant influence on the fluorescent property of poly(thioether)s. This is because the integral structure of linear polymer chain is likely to remain unchanged. ^1^H-NMR data indicates that the thiol-alkyne click reaction is a simple and an efficient approach to prepare poly(thioether)s with a special functional group by simply changing the molar ratio. Further, the remained end functional group could be used for further modification.

Theoretically, the reaction with a molar ratio of 1:1 is likely to react completely and maximize the molecular weight of polymers. However, the obtained polymers in this situation did not have reactive sites. When one monomer is excessive, the non-stoichiometric alkyne-to-dithiol molar ratio may result in a handful of functional groups in chain end of polymers. The polymers can possess short polymer chains and low molecular weight. In addition, these maintained functional groups could be used for post-functionalization and naturally various polymers with interesting properties. Carbon-carbon double bonds were maintained in chain end when the molar ratio was 1:0.8, and unreacted sulfhydryl groups were maintained when the molar ratio was 1:1.2. Moreover, the excessive dithiol can lead to the polymers possessing shorter chain lengths than excessive alkyne. Therefore, among the three different molar ratios, the sequence of their molecular weight may be as follows: *M***_w (_**_1:1)_ > *M***_w (_**_1:0.8)_ > *M***_w (_**_1:1.2)_. This speculation has been confirmed by the test outcome. The molecular weight of polymers was measured by GPC and the detailed data have been depicted in Table 1.

The thiol-alkyne reaction proceeded via radical step-growth involving two successive addition processes. The first addition generates an intermediate that contains alkenyl group, and subsequent thiol-alkene reaction yields poly(thioether)s. When one monomer is excessive, the non-stoichiometric alkyne-to-dithiol molar ratio can result in a handful of functional groups in the chain end of polymers. Poly(thioether)s containing alkenyl (**P1** and **P2**) were selected to further modification. A series of functionalized poly(thioether)s (**P1-1**, **P1-2**, **P1-3**, **P2-1**, **P2-2**, and **P2-3**) were synthesized with different sulfhydryl compounds via thiol-alkene click reaction (Scheme 1), and a detailed synthetic procedure was described in the experimental section. The modified sulfhydryl compound can bring new and interesting properties to poly(thioether)s, although the number of modified functional groups is very small.

### 3.2. Fluorescent Properties

Poly(thioether)s (**P1**, **P2**, **P3**, **P4**, **P5**, and **P6**) and functionalized poly(thioether)s (**P1-1**, **P1-2**, **P1-3**, **P2-1**, **P2-2**, and **P2-3**) emit a blue color in the solution and their emission spectra excited by 330 nm as shown in Figure 2. **P1**, **P3**, and **P5** were synthesized with different molar ratios between TMSA and EDT, their fluorescence intensity (Figure 2a) decreased sharply as the molar ratio of dithiol increased. While **P2**, **P4**, and **P6** were synthesized with different molar ratios between TMSA and DODT, their fluorescence intensity (Figure 2b) slightly decreased as the molar ratio of dithiol increased. The unconventional fluorescence can be attributed to the aggregation of long pair electrons of heteroatom (O atom and S atom). Meanwhile, the coordination bonds of O → Si and S → Si were both conducive to cause strong blue photoluminescence in poly(thioether)s [23,24]. The aggregation and coordination in poly(thioether)s containing the O atom (**P2**, **P4**, and **P6**) were stronger than poly(thioether)s without the O atom (**P1**, **P3**, and **P5**). The interaction between heteroatom and silicon atom resulted in the significant difference of fluorescence intensity among poly(thioether)s (**P1**, **P3**, and **P5**) and negligible difference of fluorescence intensity among poly(thioether)s (**P2**, **P4**, and **P6**).

The introduction of sulfhydryl compounds can bring new properties to poly(thioether)s (**P1** and **P2**). The end modified poly(thioether)s demonstrated different trends of fluorescent intensity. The functionalized poly(thioether)s (**P1-1**, **P1-2**, and **P1-3**) based on **P1** showed obvious fluorescence quenching after the end-modification (Figure 2c). The introduction of three sulfhydryl compounds all resulted in the decreasing of fluorescence intensity, especially to methyl-mercaptopropionate. The functionalized poly(thioether)s (**P2-1**, **P2-2**, and **P2-3**) based on **P2** showed slight fluorescence enhancement after the end-modification (Figure 2d). The introduction of three sulfhydryl compounds did not show a significant difference to the fluorescence intensity of **P1**. The difference between two types of functionalized poly(thioether)s indicated that the end modified functional groups had an obvious influence on the fluorescent property of poly(thioether)s, although the number of modified functional groups was very small. The different effect of the end modified groups to **P1** and **P2** could be attributed to the structure of the polymer chain. The oxygen atom in the main chain enhanced the structural stability of **P2** and weakened the influence of the end modified groups on the fluorescent properties of poly(thioether)s. The above results indicated that the fluorescent properties of functionalized poly(thioether)s were influenced by the end modified groups and the structure of the polymer chain.

### 3.3. Ion Detection and Cell Imaging

As one types of most important bioactive substances in biological systems, metal cations are vital to the stability and effectivity of the immune system. To study the responsiveness of poly(thioether)s to various cations, a series of chloride salts (Al^3+^, Ba^2+^, Ca^2+^, Co^2+^, Cu^2+^, Fe^2+^, Fe^3+^, Mg^2+^, Ni^2+^, and Sn^2+^) were added to the solutions of poly(thioether)s (**P1** and **P2**) and their end-modified derivatives. As shown in Figure 3a,e, the fluorescence intensity of **P1** and **P2** both decreased at different levels after adding metal ions, and especially for Fe^2+^ and Fe^3+^. This behavior of fluorescence quenching of poly(thioether)s after adding metal ions can be attributed to the coordination between heteroatom (O atom and S atom) and metal ion, especially for Fe^3+^. This coordination destroyed the interaction between heteroatom and Si atom, and subsequently decreased the aggregation extent of the long pair electrons of heteroatom. These two factors described above resulted in the fluorescence quenching of poly(thioether)s.

In addition, the end-modified with different sulfhydryl compounds resulted in significant differences between their responsiveness to Fe^2+^ and Fe^3+^. The introduction of methyl 3-mercaptopropionate did not bring obvious changes of **P1-1** to iron ions (Fe^2+^ and Fe^3+^) compared to **P1** (Figure 3b), while **P2-1** had negligible selectivity to Fe^2+^ and Fe^3+^ compared to **P2** (Figure 3f). However, the introduction of furfuryl mercaptan and trimethoxysilylpropanethiol significantly enhanced the selective responses of **P1** and **P2** derivates to Fe^3+^. It can be observed that **P1** derivates (Figure 3c,d) and **P2** derivates (Figure 3g,h) all exhibited good selective to Fe^3+^. The above results suggested that end-modified functional groups with a very small amount could obviously change the fluorescent properties of poly(thioether)s.

Regarding the good selectivity of functionalized poly(thioether)s to Fe^3+^, this study further explored their fluorescence behaviors in living cells. First, the HeLa cells were stained with poly(thioether)s (**P1** to **P6**) for 20 min at 37 °C and washed with PBS solution. Their confocal fluorescence images were observed and depicted in Figure 4. The images show visible blue fluorescence in the dark filed and mainly distributed in the cytoplasm. The distribution was due to the way the poly(thioether)s entered into the cell. The polymers enter the cell by simple diffusion. The blue fluorescence of poly(thioether)s belonged to unconventional fluorescence because rigid π-conjugated structures and heterocycle structures did not exist in the obtained poly(thioether)s. Furthermore, silicon atoms in the polymer chain could enhance their fluorescent properties due to their empty 3d orbital [25]. The fluorescence intensity of poly(thioether)s containing the O atom (**P2**, **P4** and **P6**) in cells were stronger than poly(thioether)s without containing the O atom (**P1**, **P3** and **P5**). This may relate to the structure of the polymer chain. The O atom in the main chain enhanced the coordination in polymers, therefore poly(thioether)s containing O atom exhibited stronger fluorescence in living cells. In addition, it can be observed that some poly(thioether)s absorbed on the surface of the cells in the bright filed. The results suggest that poly(thioether)s could easily enter living cells and could be absorbed on the surface of the cells.

To further investigate the responsiveness of poly(thioether)s to Fe^3+^ in living cells, **P1** and **P2** derivates were selected as a sample and their fluorescent quenching behaviors were observed in situ under confocal microscopy. HeLa cells were stained only with poly(thioether)s and their cell images were observed under confocal microscopy. The responsiveness of poly(thioether)s to Fe^3+^ in HeLa cells were measured by successively adding Fe^3+^ ions. The fluorescence intensity of HeLa cells in the blue field was calculated by confocal microscopy. Confocal fluorescence images of **P1** and their derivates were observed as shown in Figure 5. With the increasing Fe^3+^ concentrations, the fluorescence intensity of **P1** remained roughly unchanged. Meanwhile, the end-modification with different sulfhydryl compounds brought obvious responsiveness to Fe^3+^. **P1-1** showed slightly fluorescence quenching after adding successive concentrations of Fe^3+^. **P1-2** and **P1-3** both exhibited obvious fluorescence quenching with the increasing Fe^3+^ ions, especially for **P1-3**. Similarly, the fluorescence intensity of **P2** remained roughly the same with the increasing Fe^3+^ concentrations (Figure 6). Three types of end-modified poly(thioether)s show a similar trend in the change of fluorescence quenching.

### 3.4. Thermogravimetric Analysis (TGA)

To evaluate the thermal stability of poly(thioether)s, **P1** and **P2** were selected and their thermodynamics properties were measured by TGA analysis at 10 °C·min^−1^ in N_2_. The TGA data indicated that the poly(thioether)s exhibited high thermal stability (Figure 7). The thermal decomposition temperature (*T*_d_) of **P1** and **P2** were at approximately 290 °C and 320 °C, respectively. **P2** possessed higher *T*_d_ than **P1**, which indicated that the presence of oxygen atoms in the polymer chain increased the thermal performance of poly(thioether)s. In addition, **P1** exhibited two degradation steps. The first weight loss step of **P1** before 250 °C could be ascribed to the sublimation of low molecular weight of poly(thioether)s. The second weight loss of **P1** can be attributed to the decomposition and sublimation of the main chain. The high *T*_d_ of **P1** and **P2** suggested that the good thermal stability of poly(thioether)s, which was prepared by simple thiol-alkyne click reaction at room temperature.

## 4. Conclusions

A series of silane-based poly(thioether)s and functionalized poly(thioether)s were synthesized via successive thiol click reaction. These poly(thioether)s exhibited excellent fluorescent properties in solutions and showed visible blue fluorescence in living cells. The responsiveness of poly(thioether)s to metal ions can be adjusted by end-modifying with different sulfhydryl compounds. The functionalized poly(thioether)s showed high selectivity to Fe^3+^ detection and still exhibited responsiveness to Fe^3+^ ions in living cells. The synthetic route of poly(thioether)s described herein could be extended to synthesize novel organosilicone fluorescent materials, and their application in ion detection and cell imaging can extend the application field of poly(thioether)s.

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
