# Peer review of "Synthesis of Silane-Based Poly(thioether) via Successive Click Reaction and Their Applications in Ion Detection and Cell Imaging"

_polymers, 2019, doi:10.3390/polym11081235_

Round 1

Reviewer 1 Report

The manuscript by Gou et al. describes the synthesis of silane containing poly(thioether) and some their functionalized forms, through progressive thiol click reactions. The polymers obtained exhibited unconventional fluorescence properties, making them potentially useful for the application in cell imaging. The following minor concerns have to be considered prior publication on Polymers.

General comments.

1)  The Fig. 1, should report the position of the other peaks described in the text. Therefore, the position of the peaks at 1.7 and 1.35 ppm should be notified in the picture.

2)  The quantification data reported in Figs. 5 and 6 should include a statistical analysis. To this aim the fluorescence intensity should be read in different observation fields and mediated in order to determine the statistical significance of the variation(s) through the p value.

3)  Some methods used are not sufficiently described in terms of procedure details. For instance GPC is not defined as abbreviation and the details of the procedure is missing.

Language editing.

Line 146: change “with” with “using”.

Line 204: change “intension” with “intensity”.

Lines 220-221: change the legend in “Fluorescence spectra of poly(thioether)s P1, P3 and P5 (a), P2, P4 and P6 (b) and functionalized poly(thioether)s P1-1 to P1-3 (c) and P2-1 to P2-3 (d) at 0.2 mg/mL final concentration.” Furthermore, additional information should be provided regarding either the buffer used and the slits used in the fluorescence measurements.

Lines 244-245: additional information should be provided regarding either the buffer used and the slits used in the fluorescence measurements.

Line 254: change “observed” with “observe”.

Line 260: delete “derivatives”.

Line 283: change “indicating” with “indicates”.

Reviewer 2 Report

The paper by Gou Z. et al. describes the synthesis of a  series  of poly(thioether)s containing  silicon  atom  with  unconventional  fluorescence  via  successive  thiol  click  reaction. The exhibition of blue  fluorescence despite the lack of rigid  π- 10 conjugated  structures is quite interesting. However, although the synthesis is quite well described, more emphasis should be given on the fluorescent properties. So I suggest the publication in Polymers after revision. Here are some suggestions to the authros and corrections:

1- Abstract line 12: ‘obvious’ should be avoided. Probably visible can be used.

2- Introduction, 1st paragraph: this part can be improved. For instance, the authors mention bioimaging but this property is not mentioned in the 1st sentence where the properties are presented.

3- Introduction: I think that the introduction is only focused on the synthesis. I believe that the authors should enrich the introduction by adding a paragraph describing fluorescent imaging of cells or other systems used in biology. As far as they insist on these properties and they show results on that, the advantages of their method compared to other methods for staining with fluorescent probes should be underlined. Here, several references can be added. Here are some examples: 1- Nat. Protoc. 2012, 7 (7), 1311-1326, 2- ACS Omega 2019, 4, 10485−10493. Same argument for the responsiveness to Fe. Without referring to this at all, we come across it at only the final paragraph.

4- Materials and experiments: The procedure that the authors followed for ion detection and cell imaging as well as the polymer solutions preparation for the fluorescent studies are not mentioned at all. This part of the paper should be improved a lot.

5- Figure 1 caption: 1 should be a superscript.

6- Line 200: ‘results’ instead or ‘result’.

7- Lines 200-203: There is no verb in this sentence.

8- Lines 216-218: English should be improved.

9- Line 223: English should be improved. ‘Type’ instead of ‘types’ for instance.

10- Figure 4: This figure should be described in the main text so as people to understand what they see. How the cells were stained? In which form the polymers were introduced to the cells?

11- Line 260: ‘Investigate’ instead of ‘investigated’

12- Line 263: ‘concentration of Fe3+ solution was calculated by computer’. What this means? What kind of calculations?

13- Line 279: ‘evaluated’ replace with ‘evaluate’. Several similar mistakes should be corrected in the text.
